# Protein Acetylation Going Viral: Implications in Antiviral Immunity and Viral Infection

**DOI:** 10.3390/ijms231911308

**Published:** 2022-09-25

**Authors:** Minfei Xue, Tingting Feng, Zhiqiang Chen, Yongdong Yan, Zhengrong Chen, Jianfeng Dai

**Affiliations:** 1Department of Respiratory Medicine, Children’s Hospital of Soochow University, Soochow University, Suzhou 215025, China; 2Jiangsu Key Laboratory of Infection and Immunity, Institute of Biology and Medical Sciences, Soochow University, Suzhou 215123, China

**Keywords:** acetylation, acetyltransferases, deacetylases, viral infection, antiviral immunity

## Abstract

During viral infection, both host and viral proteins undergo post-translational modifications (PTMs), including phosphorylation, ubiquitination, methylation, and acetylation, which play critical roles in viral replication, pathogenesis, and host antiviral responses. Protein acetylation is one of the most important PTMs and is catalyzed by a series of acetyltransferases that divert acetyl groups from acetylated molecules to specific amino acid residues of substrates, affecting chromatin structure, transcription, and signal transduction, thereby participating in the cell cycle as well as in metabolic and other cellular processes. Acetylation of host and viral proteins has emerging roles in the processes of virus adsorption, invasion, synthesis, assembly, and release as well as in host antiviral responses. Methods to study protein acetylation have been gradually optimized in recent decades, providing new opportunities to investigate acetylation during viral infection. This review summarizes the classification of protein acetylation and the standard methods used to map this modification, with an emphasis on viral and host protein acetylation during viral infection.

## 1. Background

Acetylation is a widespread post-translational modification (PTM) of proteins that plays a critical role in the physiological and pathological processes of cells. Protein acetylation modulates enzyme activity, affects chromatin structure, regulates transcription, guides protein localization, and participates in protein–protein interactions and cellular metabolism. The emerging roles of PTMs involving host and viral proteins are critical for viral replication and pathogenesis. This review summarizes the current knowledge regarding protein acetylation during viral infection and discusses the importance of acetylation in viral replication and antiviral immunity.

## 2. Classification of Protein Acetylation

Acetylation is catalyzed by a series of acetyltransferases that transfer acetyl groups (-CO-CH_3_) from acetylated molecules (including acetyl-coenzyme A (Ac-CoA)) to specific amino acid residues of protein substrates [1]. Three major forms of acetylation have been described based on the sites of protein acetylation: N-terminal acetylation (also called Nα-acetylation or N-ter acetylation), lysine acetylation (Nε-acetylation or K-acetylation (KAc)), and O-acetylation.

### 2.1. N-ter Acetylation

N-ter acetylation, discovered by Kozo Narita in 1958, is an irreversible modification that occurs at the α-amino group of the protein skeleton (initial methionine (iMet) or at the penultimate amino acid after the cleavage of methionine [2]. Although N-ter acetylation is rare in prokaryotes, it is one of the most common PTMs in eukaryotes. N-ter acetylation occurs co-translationally in more than half of yeast proteins and in almost 90% of human proteins [3].

In eukaryotes, N-ter acetylation is catalyzed by N-acetyl transferase (NAT) and six NAT multiprotein complexes (NAT A-F), which transfer acetyl groups from Ac-CoA to the Nα-amino group of the N-terminal residue [4]. NAT complexes acetylate most human proteins and have different subunit compositions and substrate specificities. For example, NatA-E binds to ribosomes in the cytosol, whereas NatF binds to the Golgi apparatus [5]. In human cells, NatA, NatB, and NatC are jointly responsible for almost 80% of N-ter acetylation reactions [6]. NatA consists of the catalytic subunit Naa10 (Ard1) and the auxiliary subunit Naa15 (Nat1) [7]. However, Naa10 can also participate in N-ter acetylation independently of Naa15 [8]. NatB consists of the catalytic subunit Naa20 (Nat3) and the auxiliary subunit Naa25 (Mdm20) [9]. NatC is assembled from the catalytic subunit Naa30 (Mak3) and two auxiliary subunits: Naa35 (Mak10) and Naa38 (Mak31) [10]. The monosubunit enzymes NatD, NatE, and NatF only have one catalytic subunit: Naa40, Naa50, and Naa60, respectively [5,8,11].

The substrate specificity of NAT is determined by the first two amino acid residues. After iMet excision, NatA can acetylate S-, A-, G-, T-, V-, and C-termini [3]. NatB adds an acetyl group to the retained iMet, followed by acidic or hydrophilic amino acid residues, including D, E, N, or Q [12]. NatC, NatE, and NatF have overlapping substrate specificities, acetylating iMet, followed by amphipathic and hydrophobic amino acid residues, including A, K, I, M, V, L, F, and Y [5,8,13]. Histones H2A and H4 are the only two substrates of NatD that adds an acetyl group to the S- of S-G- residues [14].

The addition of acetyl groups can neutralize the positive charge of the N-ter residues and that can change the size of the modified amino acids and the local hydrophobicity of proteins. The biological functions of N-ter acetylation are not yet fully understood, but they are involved in modulating protein degradation, stability, protein–membrane/protein–protein interactions, and the inhibition of endoplasmic reticulum translocation [11]. In addition, N-ter acetylation is associated with several pathological processes, including tumor development and neurodegenerative diseases.

### 2.2. KAc

KAc adds acetyl groups to the side chains of the lysine residues of the substrate, and this process is reversible. Lysine acetyltransferases (KATs) and lysine deacetylases (KDACs) regulate KAc modification levels. These two groups of enzymes are sometimes called histone acetyltransferases (HATs) and deacetylases (HDACs) because histones were identified as their main substrates before discovering KAc on other cytoplasmic and nuclear proteins.

In human and mouse cells, KATs are divided into three major families according to their structural and sequence similarities (Table 1) [15,16,17,18]: general control of amino acid synthesis protein 5 (GCN5)-related N-acetyltransferases (GNATs), the p300/CBP (E1A-binding protein p300/CREB-binding protein) family, and the MYST (histone acetyltransferase) family. In addition, some KATs are not included in these three families, including protein fusion-2 (FUS2)/NAT6, embryo brain specific protein (EBS)/NAT9, sister chromatid cohesion N-acetyltransferase 1 (ESCO1), and sister chromatid cohesion N-acetyltransferase 2 (ESCO2) [11,19]. KDACs are grouped into two functional families (Table 1) [20,21,22,23]: zinc-dependent and NAD+-dependent deacetylases. In humans, KDACs are divided into four major subclasses. Among them, Class Ⅰ (KDAC1/KDAC2/KDAC3/KDAC8), Class Ⅱa (KDAC4/KDAC5/KDAC7/KDAC9), Class Ⅱb (KDAC6/KDAC10), and Class IV (KDAC11) are zinc-dependent deacetylases, while Class III (sirtuin 1-7 (SIRT1-7)) is an NAD+-dependent deacetylase [19].

As described above, the addition of acetyl groups can neutralize the positive charge of lysine residues, which changes the size of the modified amino acids as well as the local hydrophobicity of proteins [24]. This mechanism renders KAc the most significant PTM in cell signaling and metabolism. KAc can regulate the properties and functions of a variety of proteins, including DNA–protein interactions, protein stability, transcriptional activity, protein–protein interactions, metabolic regulation, cell cycle regulation, differentiation, migration, and apoptosis [25]. An imbalance in KAc is associated with several diseases, including cancer, diabetes, and neurodegeneration.

### 2.3. O-acetylation

O-acetylation is a reversible modification that adds acetyl groups to the hydroxyl side chain of serine or threonine. It has only been detected in a small number of eukaryotes. Initial reports of O-acetylation have shown that this type of PTM plays an important role in regulating biological signaling pathways (e.g., mitogen-activated protein kinases (MAPKs)) after infection with a bacterium, *Yersinia pestis*. Yersinia outer protein J (YopJ) acts as an acetyltransferase and catalyzes a series of reactions. During *Yersinia pestis* infection, O-acetylation of serine and threonine in the MAPK kinases (MKKs) and Iκ-κB kinases (IKKs) inhibits the MAPK and nuclear factor kappa B subunit (NF-κB) pathways, thereby suppressing innate immunity [26]. This is a rare event, and there have been no relevant examples involving viral infections. Recently, the O-acetylation of serine and threonine of histone H3 was demonstrated and was found to be highly conserved from yeast to humans. O-acetylation on serine 10 of histone H3 may be involved in the cell cycle and cellular pluripotency [27]. However, the occurrence of O-acetylation is relatively rare.

## 3. Acetylation of Host Proteins during Viral Infection

Viruses rely on host cell machinery to complete their genome replication, transcription, translation, packaging, and release. At the same time, they can also affect critical physiological pathways in host cells, including transcription, translation, immunity, and apoptosis. After translation, both host and viral proteins undergo PTMs to perform different biological functions (Table 2 and Table 3).

### 3.1. Proteins Associated with IFN Production

Interferon (IFN), produced after the activation of Toll-like receptors (TLRs), retinoic acid-inducible gene I (RIG-I)-like receptors (RLRs), and the cyclic GMP-AMP synthase/stimulator of interferon response cGAMP interactor (cGAS/STING) pathway, plays an essential role in the antiviral process. Several key factors can be acetylated and affect antiviral effects (Figure 1). For example, the K909 site of RIG-I is acetylated in resting Huh7 and HEK293 cells, and deacetylation of the RIG-I C-terminal region regulates its ability to sense viral RNAs. After induction with hepatitis C virus (HCV) or vesicular stomatitis virus (VSV), HDAC6 rapidly deacetylates RIG-I, which is essential for RIG-I activation and downstream IFN-β and proinflammatory cytokine production [28,29]. Similarly, during influenza A virus (IAV) infection, HDAC6 deacetylates the RIG-I C-terminal region to enhance RNA sensing, prevents viral trafficking to the membrane or RNA Pol subunit PA, and inhibits viral replication [28,30,31].

OTU deubiquitinase 3 (OTUD3) is an acetylation-dependent deubiquitinase whose catalytic activity is dependent on the acetylation of K129. OTUD3 directly hydrolyzes the K63-linked polyubiquitination of mitochondrial antiviral signaling protein (MAVS) to antagonize RLR signaling and to shut off innate antiviral immune responses. However, SIRT1 can remove K129 acetylation involving OTUD3 and can rapidly inactivate the latter, which induces a timely innate antiviral response [32]. Li et al. found that during viral infection, HDAC9 deacetylates the lysine residues of TANK-binding kinase 1 (TBK1), activating the phosphorylation of TBK1, enhancing its kinase activity, and leading to the increased induction of type I IFNs. Moreover, K241 deacetylation is critical for TBK1 kinase activity [33].

When triggered by an infection, interferon regulatory factor 3 (IRF3) is phosphorylated to form a homodimer. Acetyltransferase CBP/p300 binds to the IRF3 homodimer through the Q-rich domain, promoting the binding activity of IRF3 homodimers to DNA, resulting in IFN-β transcriptional activation [34]. The K92 in the DNA-binding domain of interferon regulatory factor 7 (IRF7) is highly conserved, and its mutation leads to the abolition of its DNA-binding ability. Caillaud et al. confirmed that p300/CBP-associated factor (PCAF) and GCN5 can acetylate IRF7 at K92 and negatively regulate IRF7 DNA-binding activity. When K92 cannot be acetylated owing to changes in the surrounding amino acid context, the DNA-binding capacity of IRF7 increases [35].

Song et al. found that KAT5 can acetylate multiple lysine residues in the N-terminal domain of the DNA sensor cGAS, increase the DNA-binding affinity of cGAS to viral DNA, and initiate a MITA/STING-dependent innate antiviral immune response to DNA viruses, including herpes simplex virus 1 (HSV-1) [36]. Similarly, two major nuclear localization motifs (NLS) of the interferon-inducible protein interferon gamma inducible protein 16 (IFI16) are acetylated at K99 and K128 by p300. Acetylation of IFI16 regulates its subcellular localization and initiates pro-inflammatory responses to limit viral replication and transmission during HSV-1 and vaccinia virus infection [37].

NF-κB is also activated in the IFN signaling pathway, producing inflammatory cytokines that are involved in immune responses. Two subunits of NF-κB, p50 and p65, are acetylated by CBP/p300. Acetylation directly regulates a variety of NF-κB functions, including transcriptional activation, DNA-binding ability, IκBα assembly, and subcellular localization. Chen et al. demonstrated that CBP/p300 is crucial for p65 acetylation at amino acid residues K218, K221, and K310. K221 acetylation enhances p65′s DNA-binding ability and impairs IκBα assembly, whereas K310 acetylation is essential for its complete transcriptional activity [38]. The acetylated form of p65 can be deacetylated by HDAC3, which promotes IκBα assembly and the rapid nuclear export of the deacetylated NF-κB complex. This replenishes the NF-κB/IκBα complex in the cytoplasmic pool in preparation for the next NF-κB-induced stimulation [39].

### 3.2. Proteins Related to IFN Downstream Signaling Pathway

IFN does not act directly on viruses; instead, it binds to IFN receptors and activates downstream signaling pathways to activate the transcription of genes related to antiviral immunity, inflammation, and apoptosis. Several key IFN downstream molecules can be acetylated and display different activities (Figure 1). CBP acetylates interferon-α receptor 2 (IFNAR2) on K399 by binding to a specific region of IFNAR2, where two adjacent proline boxes carry phosphorylated S364 and S384 residues. Interferon regulatory factor 9 (IRF9) and signal transducer and activator of transcription 1 and 2 (STAT1 and STAT2) are acetylated by CBP in their DNA-binding domains, facilitating the transcriptional activation of interferon-induced promoters [40]. Similarly, p300-mediated K685 acetylation modulates signal transducer and activator of transcription 3 (STAT3) dimerization, promoting nuclear accumulation and the transcriptional activation of STAT3 after cytokine-induced signaling [41]. Additionally, it has been shown that interferon-stimulated gene 15 (ISG15) extensively conjugates IFN-induced ISGs and viral proteins, mediating their degradation during viral infection. HDAC6 binds to the C-terminal LRLRGG of ISG15 by interacting with ISG15 and ISG15-linked proteins, promoting selective autophagic degradation [42,43,44].

### 3.3. Other Host Proteins Related to Viral Infection

The acetylation or deacetylation of tubulin also plays an important role during viral infection. Szulc-Dąbrowska et al. found that when dendritic cells and macrophages were infected with Ectromelia virus (ECTV), α-tubulin acetylation increases, which stabilizes microtubules and reduces their fluidity, contributing to viral replication and virus particle release [45]. Husain et al. demonstrated that the downregulation of Rho GTPase-mediated tubulin deacetylase and HDAC6 activity by specific inhibitors or siRNA increases α-tubulin acetylation in IAV-infected cells, which promotes virion release [46]. Microtubules (MTs) serve as specialized tracks for vesicle and macromolecule transport, and their formation is regulated by end-binding protein (EB1). During the early stages of human immunodeficiency virus (HIV-1) infection, binding between ENV/GP120 and CD4 induces the acetylation of α-tubulin and enhances the stability of MTs to promote HIV infection. EB1 depletion or expression of an EB1 carboxy-terminal fragment can inhibit MT stability and inhibit early infection. Furthermore, HDAC6 deacetylates tubulin and reduces MT stability to inhibit HIV infection [47].

During IAV infection, the HDAC inhibitor MC1568 inhibits the activity of HDAC6/8, leading to the increased acetylation of heat shock protein 90 (HSP90), which subsequently reduces the nuclear accumulation of viral polymerases, thereby attenuating viral replication [48]. Munoz-Fontela et al. found that p53 acetylation at K379 transactivates pro-apoptotic and IFN-stimulated genes (interferon regulatory factor 5 (IRF5) and IRF9) in mouse embryonic fibroblasts (MEFs) during HSV-1 and VSV infection [49]. Murray-Nerger et al. verified that the K134 acetylation of lamin B1 (LMNB1) is a molecular toggle that controls nuclear periphery stability, cell cycle progression, and DNA repair. During HSV-1 infection, LMNB1 acetylation prevents lamina disruption and inhibits viral production [50].

The matrix protein of Ebola virus (EBOV), VP40, is critical for virus budding. Zhang et al. found that p300 mediates acetylation of the conserved K667 site of E3 ligase neural precursor cells expressed developmentally downregulated protein 4 (NEDD4), promotes NEDD4-VP40 interaction, and enhances NEDD4 E3 ligase activity. NEDD4 acetylation is crucial for VP40 ubiquitination and subsequent viral budding. Moreover, the viral load of EBOV is significantly reduced in p300-knockout cell lines, indicating an important role of p300-mediated NEDD4 acetylation in the EBOV life cycle [51].

Jie et al. discovered that the regulation of H3K9 histone acetylation may be important for memory impairment induced by Borna disease virus 1 (BoDV-1). BoDV-1 infection significantly reduces H3K9 histone acetylation levels in hippocampal neurons and inhibits the transcription of synaptic genes, including postsynaptic density 95 (PSD95) and brain-derived neurotrophic factor (BDNF). Suberanilohydroxamic acid (SAHA), an HDAC inhibitor, can mitigate BoDV-1-derived negative effects [52].

When hepatitis B virus (HBV) infects hepatocytes, covalently closed circular DNA (cccDNA) is the template for the transcription of all viral mRNAs and is assembled into minichromosomes by histones and nonhistone proteins. HBx is the only regulatory protein encoded by HBV that binds to cccDNA and that modulates viral replication [53]. HBx recruits p300/CBP to cccDNA, mediates the acetylation of H3 and H4, and activates transcription [54]. HBx can also interact directly with HDAC1 and SIRT1 to hypoacetylate cccDNA-bound histones, mediating IFN-α-mediated cccDNA suppression and low levels of transcription [55,56].

**Table 2 ijms-23-11308-t002:** Acetylation of host proteins during viral infection.

Protein	Acetylation Site	Function	Reference
RIG-I	K909	Deacetylation is critical for RIG-I activation in vivo and the production of IFN-β and pro-inflammatory cytokines	[28,29]
OTUD3	K129	Hydrolyzes Lys63-linked poly-ubiquitination of MAVS to antagonize RLR signaling and shuts off innate antiviral immune response	[32]
TBK1	K241	Deacetylation of TBK1 activates the phosphorylation of TBK1, enhances its kinase activity, and leads to an increased induction of type I IFNs	[33]
IRF3		Prompts the DNA-binding activity of IRF3 homodimer	[34]
IRF7	K92	Negatively modulates IRF7 DNA binding	[35]
cGAS	N-terminal domain	Has a higher affinity to viral DNA and initiates a MITA/STING-dependent innate immune response to DNA viruses	[36]
IFI16	K99, K128	Regulates subcellular localization and determines the initiation of pro-inflammatory responses and innate immune signaling when infected with DNA viruses, limiting viral replication and transmission	[37]
P65	K218, K221, K310	Regulates different NF-kB functions	[38]
IFNAR	K399	Recruits IRF9, STAT1, and STAT2	[40]
	K685	Regulates STAT3 dimerization and promotes nuclear accumulation of STAT3 and transcriptional activation following cytokine-induced signaling	[41]
ISG15	C-terminal LRLRGG	Deacetylation promotes the recycling of ISG15, targets selective autophagic degradation, and inhibits viral transmission	[42,43,44]
α-tublin		Enhances microtubule stability	[45,46]
HSP90	K294	Reduces nuclear accumulation of viral polymerases and attenuates viral replication	[48]
p53	K379	Trans-activates pro-apoptotic and IFN-stimulated genes to promote virus-induced apoptosis and activates the interferon pathway to enhance the antiviral effect	[49]
LMNB1	K134	A molecular toggle that controls nuclear periphery stability, cell cycle progression, and DNA repair; prevents lamina disruptions to inhibit virus production	[50]
NEDD4	K667	Essential for activating VP40 ubiquitination and virus budding	[51]
H3K9		Plays an important role in BoDV-1-induced memory impairment	[52]
Histone	H3, H4 subunits	Activates the transcription process	[54]

**Table 3 ijms-23-11308-t003:** Acetylation of viral proteins during viral infection.

Virus	Protein	Acetylation Site	Function	Reference
BmNPV	LEF-3, LEF-4, LEF-6, LEF-11		LEF-3 is essential for the expression of late viral genes and viral DNA replication; others to be determined	[57]
HPV	E7		Disrupts the transcriptional activation of IL-8 promoter, leading to the down-regulation of cellular immune response to infection	[58]
	E2	K111	Necessary for Topo1 recruitment to the viral origin to remove replication-inhibitory DNA supercoiling	[59]
IAV	M1	S195, S196, S207, K95	Not determined	[60]
	NP	S274, S283, S287, S326, S403, K325	Not determined	[60]
		K31, K90, K184	Regulates the interaction with vRNA, mRNA, or cRNA	[61]
		K77, K113, K229	Promotes the replication and survival of the virus	[62,63]
	NS1	K108	Promotes the replication and survival of the virus	[62,63]
	PA	K102, K104, S631	K102 acetylation is critical for PA endonuclease activity, especially for mRNA cap-binding activity; others to be determined	[60]
		K19	Enhances endonuclease activity and RNA-dependent RNA polymerase activity and affects the transcription process of the virus	[64]
HIV-1	IN	K258, K264, K266, K273	K264, K266, and K273 are critical for the integration of HIV-1 DNA into the host genome; others to be determined	[65,66]
	Tat	K28, K50, K51	Regulates the transcriptional process	[67,68,69,70]
SARS-CoV	N	K26, K389	Not determined	[71]
SARS-CoV-2	N	K61, K100, K102, K237, K248, K249, K266, K355, K374, K375, K387, K388	Not determined	[71]
HDV	S-HDAg	K72	Regulates the shuttle signal, which is critical in the life cycle of HDV	[72]
EBOV	NP	K272, K274, K281, K352, K404, K513, K617	Not determined	[73]
	VP40	K221, K224, K225, K274, K275	Not determined	[73]

## 4. Acetylation of Viral Proteins

The acetylation of viral proteins is also important in the viral life cycle (Table 3 and Figure 2). Hu et al. detected 39 KAc sites on 22 proteins of *Bombyx mori* nucleopolyhedrovirus (BmNPV) via a proteomic method. Certain late expression factors (LEFs) of BmNPV, including LEF-3, LEF-4, LEF-6, and LEF-11, exhibit high acetylation levels. LEF-3 contains up to four KAc sites and is essential for the expression of late viral genes and viral DNA replication [57].

Interactions between histone acetyltransferase PCAF and HPV protein E7 disrupt the transcriptional activation of the *IL*-*8* promoter, leading to the downregulation of the cellular immune responses to infection [58]. In addition, K111 acetylation in HPV protein E2 induced by p300 is crucial for HPV replication. The acetylation of K111 is necessary for Topo1 recruitment to the viral origin and for the removal of replication-inhibitory DNA supercoiling [59].

The IAV genome encodes 10 major viral proteins, including matrix proteins (M1 and M2), hemagglutinin (HA), neuraminidase (NA), nucleoprotein (NP), NS1, NS2, and RNA polymerase subunits PA, PB1, and PB2, together with at least seven accessory viral proteins. Using a proteomic approach, Ahmed et al. found that the M1, NP, and PA of IAV could be modified by acetylation. The M1 protein, critical for IAV assembly, virion shape, and integrity, is acetylated at S195, S196, S207, and K95. NP is acetylated at S274, S283, S287, S326, S403, and K325. PA is acetylated and methylated at K102 and K104, where differential modifications are highly conserved among 800 sequences from the 48 IAV subtypes that are currently recognized. PA can also be acetylated at S631. K102 is essential for PA endonuclease activity, especially in mRNA cap-binding [60]. Hatakeyama et al. reported that in the NP protein, which is close to the RNA-binding groove, K31 and K90 could be acetylated by PCAF and GCN5, respectively. In addition, K184 of NP is acetylated by these two acetyltransferases. The acetylation of these three lysine residues is involved in regulating their interactions with vRNA, mRNA, or cRNA. Interestingly, viral polymerase activity increases after silencing PCAF and decreases after silencing GCN5, suggesting that the acetylation of K31 and K90 has opposite effects on viral replication [61]. Hatakeyama et al. also verified that PCAF and GCN5 can acetylate K19 on the PA subunit of the IAV RNA-dependent RNA polymerase, enhancing its endonuclease and RNA-dependent RNA polymerase activity while also affecting viral transcription [64]. Ma et al. reported that the acetylation of K77, K113, and K229 on NP and of K108 on NS1 of IAV enhances the activity of these proteins and induces IFN antagonism, which promotes viral replication and survival [62,63].

Acetylation is also important in HIV-1 pathogenesis. Four lysine residues (K258, K264, K266, and K273) at the C-terminus of the IN protein can be acetylated by GCN5, and three lysine residues (K264, K266, and K273) are also modified by p300. The acetylation of K264, K266, and K273 increases the affinity between IN and DNA and promotes its DNA strand transfer ability, which is critical for the integration of reverse-transcribed HIV-1 DNA into the host genome [65,66]. Similarly, K28, K50, and K51 in Tat are acetylated, leading to increased transactivation activity of modified proteins on the viral LTR promoter. p300 acetylates K50 and promotes the separation of Tat from *TAR* RNA, an important process in the early stages of transcriptional elongation [69]. PCAF (KAT2B) enhances the binding of Tat to the related kinases cyclin-dependent kinase 9 (CDK9)/positive transcription elongation factor b (P-TEFb) by acetylating K28 [69]. Meanwhile, GCN5 (KAT2A) significantly increases the transcriptional activity of Tat via the acetylation of K50 and K51 [67]. Interestingly, Sakane et al. found that demethylation occurs during the Tat activation cycle. Lysine-specific demethylase 1 (LSD1/KDM1) activates Tat transcriptional activity in a K51-dependent manner and is a Tat K51-specific demethylase. Moreover, Tat K50 acetylation inhibits the monomethylation of Tat K51 [70]. Other studies have shown that HDAC6 mediates the deacetylation of Tat K28, which inhibits Tat-mediated transcriptional activity of the HIV-1 promoter [68].

Using liquid chromatography–mass spectrometry analyses (LC-MS/MS), Hatakeyama et al. demonstrated that the N proteins of severe acute respiratory syndrome coronavirus (SARS-CoV) and SARS-CoV-2 are acetylated by PCAF and GCN5 in vitro. The K267 and K389 of N SARS-CoV and the K61, K100, K102, K237, K248, K249, K266, K355, K374, K375, K387, and K388 of N SARS-CoV-2 are acetylated. K61 acetylation may reduce the affinity between the N protein and viral RNAs. The acetylation of K237, K248, and K249 modulates interactions between the N and M proteins. Acetylated K374 and K375 exist in the RNA-binding sites, which may affect affinity between N proteins and viral RNAs [71].

The RNA genome of the hepatitis D virus (HDV) encodes two viral nucleocapsid proteins, the small and large forms of the HDV antigen (S-HDAg and L-HDAg), both of which can be acetylated. S-HDAg is required for the viral RNA replication, and L-HDAg is involved in the assembly of viral particles. Nucleocytoplasmic shuttling of viral genomic RNA is vital for the HDV life cycle. Mu et al. demonstrated that the acetylation of S-HDAg may regulate this shuttling, and K72 has been identified as one of the acetylation sites of S-HDAg [72].

Moreover, p300/CBP and PCAF can acetylate lysine residues in the NP and VP40 of the Zaire EBOV. Acetylated NP lysine residues are located in the α-helix exposed on the surface of its VP35-binding domain. Acetylated lysine residues of VP40 are localized in a basic patch, which is a key region for interactions involving the viral envelope and nucleocapsid. NP K281, K352, K404, and K513 can be acetylated by both p300/CBP and PCAF, whereas K272 and K274 can only be acetylated by p300/CBP, and K617 can only be acetylated by PCAF. The acetylation of these sites may regulate the activity of viral RNA polymerase and interactions between NP and the viral RNA genome. In VP40, p300/CBP and PCAF acetylate K221, K224, K225, K274, and K275. K274 acetylation may regulate the interaction between VP40 and phosphatidylserine and may affect the efficiency of viral budding [73].

## 5. Methods of Studying Protein Acetylation

The identification of acetylated proteins, the distribution of acetylation sites, and the acetylation levels are the three major evaluation factors for studying the biological functions of protein acetylation. Radiolabeling and Western blotting are generally used for the detection and for the semi-quantitative analysis of targeted protein acetylation. Edman degradation, MS, and nuclear magnetic resonance spectroscopy (NMR) have been used to identify acetylation types, acetylation levels, and the distribution of acetylated sites.

### 5.1. Detection of Acetylated Proteins

Radiolabeling, which can be performed in vivo or in vitro, is used to detect protein acetylation. Generally, [^14^C]-or [^3^H]- sodium acetate-labeled proteins can be separated by one- or two-dimensional gel electrophoresis and can be detected by autoradiography or scintillation counting [74]. Radiolabeling is a sensitive method for detecting N-ter acetylation, KAc, and O-acetylation with a high level of specificity. Although this approach has been widely used, it has two major shortcomings: (1) radioactive elements are hazardous to operators and the environment; and (2) further studies cannot be performed for the identified acetylated proteins, for example, identifying the distribution of acetylation sites or determining acetylation types [19].

In addition, Western blotting can be performed to detect acetylated proteins using specific anti-acetylated protein antibodies [75]. Currently, Western blotting is the most widespread method for detecting the acetylation of lysine residues because of its high sensitivity and absence of radioactivity. However, it also has several limitations [76,77]: (1) it produces a large number of non-specific signals; (2) the distribution of acetylation sites or types cannot be determined by Western blotting; and (3) most antibodies are only used for KAc, whereas there are no suitable antibodies for the study of N-ter and O-acetylation.

### 5.2. Enrichment or Fractionation of Acetylated Proteins/Peptides

Because of the low abundance of protein acetylation, it is a considerable challenge to identify and quantify protein acetylation as well as the distribution of protein acetylation sites. Therefore, it is important to fractionate or enrich acetylated proteins prior to conducting a series of studies. Currently, the standard methods used for the fractionation and enrichment of acetylated proteins include immunoprecipitation (IP), strong cation exchange chromatography (SCX), and zwitterionic hydrophilic interaction liquid chromatography (ZIC-HILIC).

IP is an efficient method of protein enrichment and fractionation. Acetylated proteins or peptides are enriched by anti-acetylation amino acid antibodies before analysis, which can improve the accuracy and precision of trials. SCX is mainly used to enrich peptides via phosphorylation and N-ter acetylation based on their different charges in solution [78,79]. After digestion with trypsin, N-ter-acetylated proteins have a lower charge than non-acetylated proteins at low pH, so they are enriched and fractionated from digested peptides via the SCX process. ZIC-HILIC is another option for enriching and fractionating N-ter-acetylated peptides. ZIC-HILIC minimizes the co-elution of phosphorylated and N-ter acetylated peptides compared to SCX. It provides a better resolution by reducing the interference of salts or buffers required in the SCX mobile phase [19].

### 5.3. Identification, Localization, and Quantitative Analysis of Acetylated Proteins

Edman degradation, MS, and NMR are common methods used for the identification, localization, and quantitation of acetylated proteins. Edman degradation is commonly used to sequence unmodified peptides and has also been used to identify acetylated lysine. However, Edman degradation requires a large number of purified samples and can only detect 20–25 residues, which is time-consuming. Hence, this is not an optimal method for studying protein acetylation [19].

MS is an important method for the study of protein acetylation, and it can detect acetylated proteins or peptides, identify acetylated sites, and quantify acetylated proteins. In general, the analysis of protein acetylation is carried out by a continuous process that includes the digestion of proteins; the enrichment or fractionation of acetylated peptides; MS or MS/MS; and bioinformatic analysis. The quality of peptides is critical for the accuracy and precision of the MS results. Ahmed et al. improved the content and purity of peptides through methods including in-gel digestion and filter-aided sample preparation (FASP) [60].

NMR is a non-destructive technique used to study acetylation. Selective isotope labeling, high-resolution NMR, and 2-D heteronuclear NMR (^1^H-^15^N and ^1^H-^13^C NMR) are commonly used to identify acetylation [80,81,82]. However, NMR is used only for analyzing acetylated peptides, not proteins or complex mixtures, because overlapping peaks may impact the protein analysis results.

## 6. Concluding Remarks and Future Perspectives

Acetylation is a common but important PTM that modulates enzyme activity, chromatin structure, gene transcription, protein localization, protein–protein interactions, and cellular metabolism. Simultaneously, both host and viral proteins can be acetylated and play significant roles during viral infection. Thus, the characterization of additional host and viral proteins in various infection models is worthy of further study. Much research is currently focused on KAc, but relatively little research has been conducted on N-ter- and O-acetylation. Apart from the common acetylation of lysine residues, the acetylation of histidine, arginine, serine, threonine, tyrosine, and aspartic acid residues has also been reported. New technologies, including MS and NMR, have emerged to study the acetylation of proteins in addition to conventional biological experiments such as Western blotting and radiolabeling. In addition, bioinformatic tools and databases are essential for identifying the biological background of acetylation, searching for acetylation sites, and understanding relationships with other PTMs. Consequently, to study acetylation, it is necessary to effectively combine biological experiments with bioinformatic analysis to obtain more accurate conclusions.

## Figures and Tables

**Figure 1 ijms-23-11308-f001:**
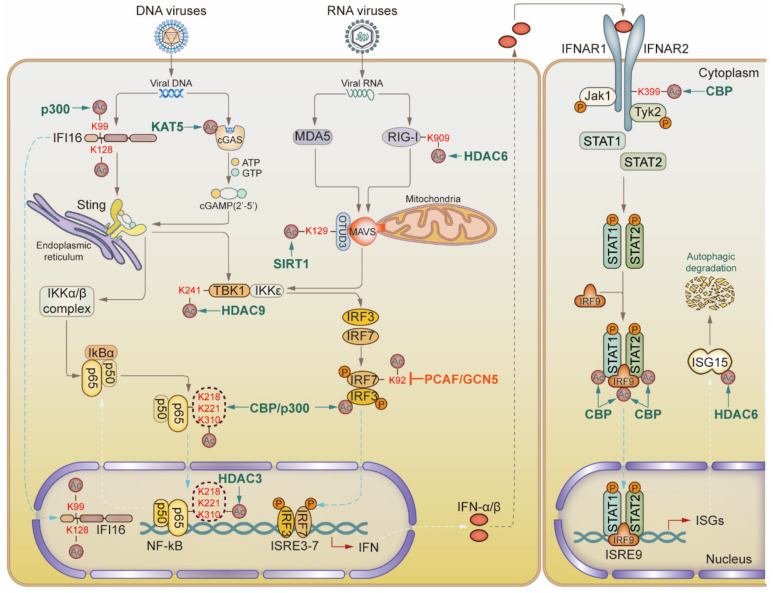
Acetylation modulates key factors of type Ⅰ interferon (IFN) pathway during viral infection. In the upstream IFN signaling pathway, molecules such as cGAS, RIG-I, MAVS, IRF3, IRF7, and NF-κB, can be acetylated or deacetylated by specific acetyltransferases or deacetylases, modulating the production of IFN. In the downstream IFN signaling pathway, CBP can acetylate IFNAR2, STAT1, and STAT2, facilitating transcriptional activation of interferon-stimulated genes.

**Figure 2 ijms-23-11308-f002:**
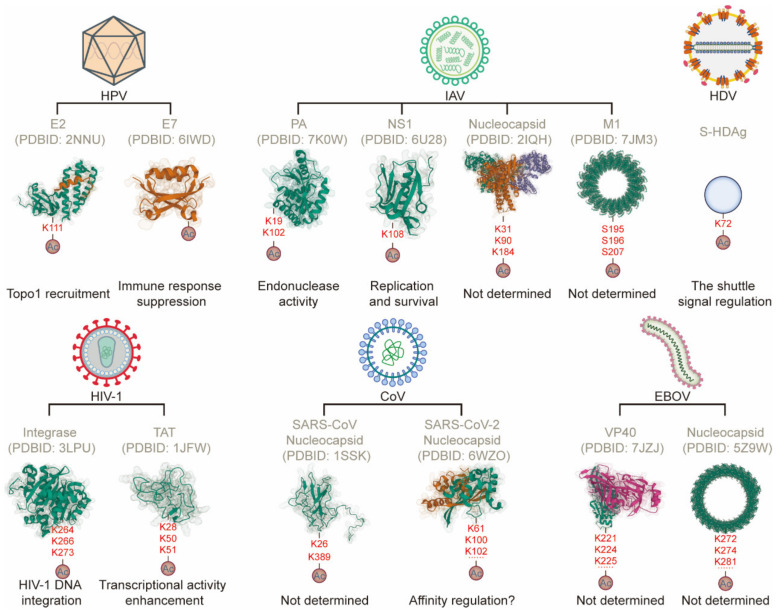
Effect of acetylation of viral proteins during viral infection. Representative acetylation of viral proteins and their function during virus–host interaction. The 3-D structures for the viral proteins were retrieved from the PDB database (PDB accession numbers 2NNU, 6IWD, 7K0W, 6U28, 2IQH, 7JM3, 3LPU, 1JFW, 1SSK, 6WZO, 7JZJ, and 5Z9W). The acetylation chains are representative of hypothetical schematics.

**Table 1 ijms-23-11308-t001:** Lysine acetyltransferases (KATs) and lysine deacetylases (KDACs).

Family	HAT/KAT	New Name	Catalytic Mechanism	Class	Member
GNAT	HAT1	KAT1	Zn^+^-dependent	I	HDAC1
	GCN5(GCN5L2)	KAT2A			HDAC2
	PCAF	KAT2B			HDAC3
	ELP3	KAT9			HDAC8
	TFIIIC90(GTF3C4)	KAT12		IIa	HDAC4
	SRC-1(NCOA1)	KAT13A			HDAC5
	SRC-3(TRAM1/NCOA3/ACTR)	KAT13B			HDAC7
	SRC-2(TIF2/GRIP1/bHLHe75/NCOA2/P160)	KAT13C			HDAC9
	CLOCK	KAT13D		IIb	HDAC6
	ACAT1				HDAC10
	ATAT1			IV	HDAC11
	ATF-2(CREB2/CREBP1)		NAD^+^-dependent	III	SIRT1
	NAT10				SIRT2
	GCN5L1(BLOC1S1)				SIRT3
p300/CBP	CBP	KAT3A			SIRT4
	p300	KAT3B			SIRT5
	TAF1(TAFⅡ250)	KAT4			SIRT6
MYST	TIP60/PLIP	KAT5			SIRT7
	MOZ/MYST3	KAT6A			
	MORF/MYST4	KAT6B			
	HBO1/MYST2	KAT7			
	MOF/MYST1	KAT8			
Others	EBS/NAT9				
	ESCO1				
	ESCO2				
	FUS2/NAT6				

Note: The references for lysine acetyltransferases (KATs) [11,15,16,17,18,19] and lysine deacetylases (KDACs) [20,21,22,23].

## Data Availability

Data sharing not applicable to this article as no data sets were generated or analyzed during the current study. Data cited in this review are published and available online or upon request from the authors of the respective publications.

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
