# Peer review of "Protein Acetylation Going Viral: Implications in Antiviral Immunity and Viral Infection"

_ijms, 2022, doi:10.3390/ijms231911308_

Round 1

Reviewer 1 Report

This manuscript by Xue et al gives a detailed and very comprehensive overview of acetylation events occurring during viral infections - including both host and viral proteins. The use of summary tables listing host and viral proteins that are acetylated was especially helpful in summarizing this level of information. A few minor points below:

Line 29 - 33: font is a different size

Line 33 - 35: this information is not required

Line 96: 'and so on' is not the best terminology to use. Either list the others that aren't included in one of the 3 families, or perhaps refer to the table

Line 121 - 125: why has Yersinia pestis been used as an example of O-acetylation when it's a bacterium (given that this is a review about acetylation during viral immunity?). I understand this is a rare event and there may not be any relevant examples in viral infections. Perhaps specify this and just mention that Yersinia pestis causes bacterial infections.

Line 412: referenced (Ahmed) is incomplete - or potentially a typo?

There are a lot of abbreviation used in this review - a list or table at the end would be helpful

Reviewer 2 Report

In this manuscript authors intended to give an overview on protein acetylation and its role in modulating virus-host interactions. Although this is not the first on this topic, an updated review is warranted for this fast-moving field. Having said that, I feel the organization of the review needs to be improved and English language needs to be polished. I suggest the following points for authors to consider:

1.       Sections 3, 4, 5 discuss three forms of acetylation. I think they should be listed as subsections 2.1, 2.2, and 2.3 to form a complete section.

2.       Section 6 is redundant and not absolutely necessary. Instead, this should be a section on “Acetylation of host proteins during viral infection” with “Proteins associated with IFN production”, “Proteins related to IFN downstream signaling pathway”, and “Other host proteins related to viral infection” being the subsections.

3.       Sections 10, 11, and 12 should be subsections for “Methods of studying protein acetylation”. Actually, this section could be moved ahead of the host protein and viral protein sections to become Section 3. Together with Sections 1 and 2, it is part of a general introduction on protein acetylation.

4.       A perspective and future direction section should be added before “Conclusions”. This is the opportunity for the authors to elaborate on their thoughts on whether there is any significant gap in the field, and what is to come.

5.       There are places where I have a hard time to understand what the authors really wanted to convey. Editing by a native speaker is recommended.     
